

# *Holothuriophilus trapeziformis* Nauck, 1880 (Decapoda: Pinnotheridae) from the Pacific coast of Mexico: taxonomic revision based on integrative taxonomy

Fernando Cortés-Carrasco[1], Manuel Elías-Gutiérrez[2] and María del Socorro García-Madrigal[3]

[1] Departamento de Sistemática y Ecología Acuática, El Colegio de la Frontera Sur Unidad Chetumal, Chetumal, Mexico
[2] Departamento de Sistemática y Ecología Acuática, El Colegio de la Frontera Sur, Chetumal, Quintana Roo, Mexico
[3] Laboratorio de Sistemática de Invertebrados Marinos (LABSIM), Universidad del Mar, Puerto Ángel, Oaxaca, Mexico

## ABSTRACT

**Background:** *Holothuriophilus trapeziformis* Nauck, 1880 is a holothurian-dweller pinnotherid crab representing one of the two species of the genus distributed along the Pacific coast of Mexico and Chile, respectively. While the parasitic ecological interaction with its host is well established, the morphology of the male remains unknown, and DNA information for the species is not available. Furthermore, the only morphological trait separating both species of the genus is subjective and corresponds to the presence or absence of a gap between the fingers of the chelae. Our goal is to complete and clarify the taxonomic status of *H. trapeziformis* and describe the male morphology with the use of the integrative taxonomy, providing additional characters to differentiate this species.

**Methods:** We collected new biological material in the Pacific coast of Mexico including the topotypes. We also reviewed material from national collections to integrate morphology (based on a complete and detailed description and illustration of the species using light microscopy), ecological data (based on the identification of the host and the place where it was located within the host), and the mtCOI gene information (commonly known as DNA barcode) to differentiate *Holothuriophilus trapeziformis* from other related crabs.

**Results:** This species presents marked sexual dimorphism only in the primary sexual characters. For the first time we describe morphological variability of traditionally stable characters. In addition to the gap between the fingers of the chelae, *Holothuriophilus trapeziformis* differs from *H. pacificus* (Poeppig, 1836) by their ornamentation, the shape of the male abdomen, and the gonopod. Cytocrome Oxidase 1 gene (COI) distance divergence was >3% between both *Holothuriophilus* species forming a clear clade. DNA barcoding indicates only one taxon, with a maximum divergence of 2.2%. All the specimens have the same Barcode Index Number (BIN; BOLD: ADE9974). All the hosts for *H. trapeziformis* were identified as *Holothuria* (*Halodeima*) *inornata* Semper, 1868; the presence of the crab in the host's coelomic cavity was confirmed, and for the first time we found it within the intestine. The geographical distribution is the Pacific coast of Mexico. Based on the data

Corresponding author
Manuel Elías-Gutiérrez,
melias@ecosur.mx

presented here, the taxonomic status of *Holothuriophilus trapeziformis* is now complete.

## INTRODUCTION

Pinnotherids (Crustacea: Pinnotheridae) are true decapod crabs, which show conspicuous sexual dimorphism, notably different morphological stages of development and complex ecological relationships with different invertebrates, however, they can also be found freeliving (*Schmitt, McCain & Davidson, 1973*; *Ocampo et al., 2011*; *Becker & Türkay, 2017*). Worldwide, 16 species are known to be endobiontic with sea cucumbers (*Ng & Manning, 2003*). Of these, only two have been recorded on the Pacific coast of America and they are: *Holothuriophilus trapeziformis Nauck, 1880* (type locality in Mazatlan, Mexico, associated with the sea cucumber *Holothuria* (*Halodeima*) *inornata Semper, 1868*), and *H. pacificus* (*Poeppig, 1836*) (from Talcahuano, Chile, associated with *Athyonidium chilensis* (*Semper, 1868*)) (*Garth, 1957*; *Manning, 1993*; *Honey-Escandón & Solís-Marín, 2018*). *Holothuriophilus* (*Nauck, 1880*) is diagnosed by its transversally elongated carapace, wider anterior to middle portion; its short, robust and compressed walking legs, with the dorsal margin crested; and the third maxilliped with the ischiomerus indistinguishably fused (*Garth, 1957*; *Manning, 1993*; *Ng & Manning, 2003*; *Campos, Peláez-Zárate & Solís-Marín, 2012*).

The taxonomic status of *H. trapeziformis* remains incomplete, because male morphology is unknown and the available information from female illustrations shows some inconsistencies when the carapace, Mxp3 shape, and setae patterns are compared (see *Bürger, 1895*: 380–381, pl. 9, fig. 26; *Ahyong & Ng, 2007*: 214, Figs. 20A, C; *Campos, Peláez-Zárate & Solís-Marín, 2012*: 60, figs. 2B, C). Additionally, *Holothuriophilus trapeziformis* and *H. pacificus* only differ in a gap when the fingers of the chelae closed in the latter (*Campos, Peláez-Zárate & Solís-Marín, 2012*).

*Nauck's (1880)* description of *Holothuriophilus trapeziformis* was brief, a holotype was not designed and the identity of the type host was misidentified. Moreover, the female syntypes deteriorated over time (*Bürger, 1895*; *De Man, 1887*; *Ng & Manning, 2003*). Later, *Manning (1993)*, *Ng & Manning (2003)*, and *Ahyong & Ng (2007)* examined, described and illustrated the syntype series to complete the diagnosis and designated a lectotype. However, there are inconsistencies between their illustrations and the diagnostic characters are not informative with the information available for *Holothuriophilus pacificus*.

For 84 years there were no new records of *H. trapeziformis* until *Caso (1958*, *1964*, *1965*) reported four pinnotherids, determined as *Pinnixa barnharti* (not *Pinnixa barnharti Rathbun, 1918*), associated with *Holothuria inornata Semper* and *H. kefersteinii* (Selenka) (= *H. riojai Caso, 1964*).

Thirty-four years later, *Campos, Díaz & Gamboa-Contreras (1998)* determined as *Holothuriophilus* sp. the specimens of *Caso (1964)*. More recently *Campos, Peláez-Zárate & Solís-Marín (2012)* reviewed the genus and updated the species diagnosis. Finally, *Honey-Escandón & Solís-Marín (2018)* confirmed the ecological association between *H. trapeziformis* and *Holothuria inornata*, but the specimens of *Caso (1958, 1965)* associated to *Holothuria kefersteinii* remain uncertain because they have not been included in any of these documents and their location is unknown (F. Solís-Marín, 2018, personal communication).

For *Holothuriophilus trapeziformis*, there is currently no data on any gene, while for *H. pacificus*, there is information related to the Cytocrome Oxidase 1 gene (COI) sequence for one specimen recovered from the shoreline in southern Chile (CFAD062-11; boldsystems.org). Within this context, sequencing of approximately 650 bp region of the mitochondrial Cytocrome Oxidase 1 gene (COI) has been promoted to conform a standardized DNA barcode system with the aim of being one more tool for the identification of biological species with many applications in diverse fields of knowledge (*Hebert et al., 2003*; *Hajibabaei et al., 2007*). Despite the fact that the acceptance to work with a single molecular marker as a precise character to define a species was discussed (*Will & Rubinoff, 2004*), nowadays it is considered that the COI marker is the best for decapod identification (*Spielmann et al., 2019*). The utility of DNA Barcoding (COI sequence) has been useful to delimit species of pinnotherids (*Ocampo et al., 2013*; *Perez-Miguel et al., 2019*), brachyuran larvae (*Brandão, Freire & Burton, 2016*) and other crustaceans (*Costa et al., 2007*; *Matzen da Silva et al., 2011*).

Therefore, the goal of this study is to review the taxonomy of *Holothuriophilus trapeziformis* by describing the male, examining the morphological variability in both sexes, updating the range of distribution, and establishing a baseline of mitochondrial COI gene barcode. Finally, with this information, we propose new diagnostic characters that will allow better identification of the two species of *Holothuriophilus* known in the Pacific coast of America.

## MATERIALS AND METHODS

### Morphology

Fifty-five crabs belonging to *Holothuriophilus trapeziformis* were extracted from the coelom and intestine of the sea cucumber *Holothuria* (*Halodeima*) *inornata*. These were collected in 11 locations from the Pacific coast of Mexico (Fig. 1). These localities were grouped into two arbitrary regions designated as the northern region (Sinaloa state; which include the type locality Mazatlan) and the southern region (Guerrero and Oaxaca states). Hosts were manually collected through skin and SCUBA diving at a maximum depth of 10 m. The collected material was labeled and fixed according to *Elías-Gutiérrez et al. (2018)* protocol for tissue preservation and DNA analyses. Due to the size of the specimens and the thickness of the cuticle, we injected ethanol into the body through the joints of the appendices with insulin syringes.

All biological material (Table S1) was classified and deposited in the Scientific Collection of Marine Invertebrates of the Laboratorio de Sistemática de Invertebrados
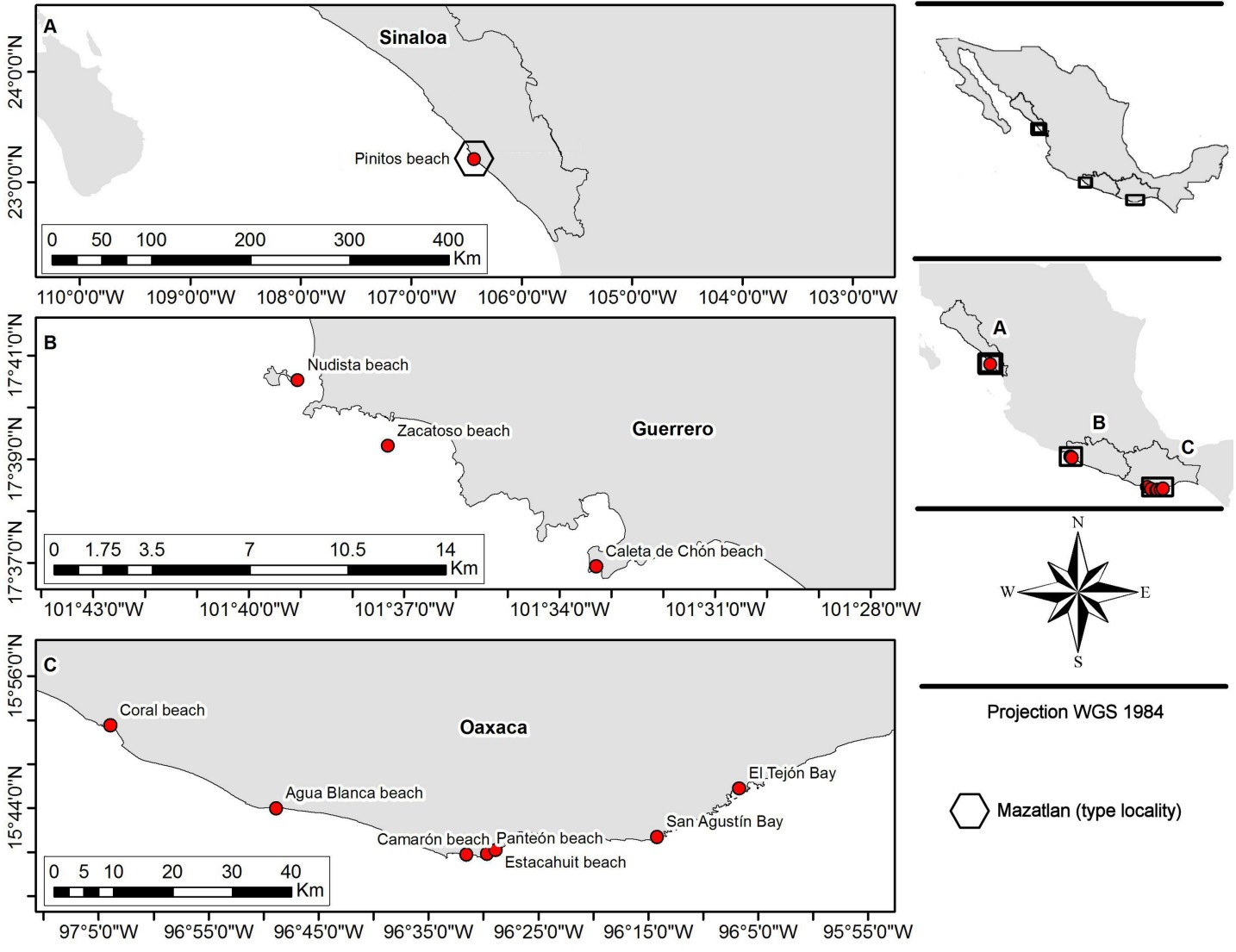

**Figure 1 Sampling sites.** (A) Northern region; type locality Mazatlan (*fide Nauck, 1880*); (B and C) southern region.

Marinos (LABSIM) at Universidad del Mar (UMAR), Oaxaca, Mexico (OAX-CC-249-11). Hosts were identified with specialized literature (*Solís-Marín et al., 2009*; *Honey-Escandón & Solís-Marín, 2018*).

For the analysis of the taxonomic status of *Holothuriophilus trapeziformis* specialized literature from *Nauck (1880)*, *Manning (1993)*, *Ng & Manning (2003)*, *Ahyong & Ng (2007)*, and *Campos, Peláez-Zárate & Solís-Marín (2012)* was reviewed. Likewise, for *H. pacificus*, *Poeppig (1836)*, *Nobili (1901)*, *Rathbun (1918)*, and *Garth (1957)* were reviewed.

The species description follows the terminology of *Manning (1993)*. We consider the gonopod terminology as first gonopod and second gonopod of the male, according to *Becker, Türkay & Brandis (2012)*. The setae terminology is based on *Garm & Watling (2013)*. Drawings were made with the help of a camera lucida and then digitalized in a

vector format. Pictures were taken with a Nikon D5100 digital camera. Measurements are given in millimeters and latitude and longitude were obtained from Google Earth$^{TM}$.

Because we were only able to obtain eight specimens (three males and five females) from the northern region (Pinitos Beach in the type locality Mazatlan, Sinaloa, Mexico), in contrast to 47 (six males and 41 females) from localities in the southern region, and due to morphological variability observed between these two regions, it was necessary to standardize the observations by using specimens in the same stage of development. Eight millimeters carapace width was the common size in both, the northern and southern regions, in males and adult ovigerous females. This size-group was used to illustrate and discuss the observed morphological variability. The specimen and the dissected appendages were mounted on a plastic clay base for standardizing the observations, and to make the drawings. For the carapace contour, the samples were mounted, so the dorsal view of the posterior margin line of the carapace was observed. For the Mxp3, we extracted it from its base to obtain both endopod and exopod and to mount it with the articles in the same perspective. The cutting edge of the fingers' chelae was cleaned of dirt to see all the teeth. The first gonopod was extracted from its base, and the setae cleaned of dirt.

Abbreviations used in the text: CL, carapace length (taken as the middle line from the frontal margin to the posterior margin of the carapace); CW, carapace width (measured in its medium-anterior portion); Mxp2, second maxilliped; Mxp3, third maxilliped; WL, walking legs 1 to 4 (thoracopods 2–5).

## DNA extraction and PCR amplification

Genomic DNA of individuals of *Holothuriophilus trapeziformis* was extracted from biological material collected in the field and some individuals from the OAX-CC-249-11 regional collection of the UMAR, using tissue from the walking legs, the chelae, or eggs from the ovigerous females. Tissues were placed into 96-well microplates with a drop of 96% ethanol, and DNA extraction was carried out following the standard glass fiber method consisting of a mix of Proteinase K with an invertebrate lysis buffer according to *Ivanova, De Waard & Hebert (2006)*. Following the DNA extraction, the PCR mixture with a final volume of 12.5 µl contained 2 µl of Hyclone ultrapure water (Thermo Fisher Scientific, Waltham, MA, USA), 6.25 µl of 10% trehalose (previously prepared: 5 g D-(+)- trehalose dihydrate (Fluka Analytical, St. Louis, MO, USA) in a total of 50 ml of molecular grade ddH$_2$O), 1.25 µl of 10X PCR Platinum Taq buffer (Invitrogen, Waltham, MA, USA), 0.625 µl of 50 µmol/L MgCl2 (Invitrogen, Waltham, MA, USA), 0.0625 µl of 10 µmol/L dNTP (KAPA Biosystems, Wilmington, MA, USA), 0.125 µl of each 10 µmol/L primer, 0.06 µl of PlatinumTaq DNA polymerase (Invitrogen, Waltham, MA, USA), and 2 µl of DNA template. All specimens were amplified with the Zooplankton primers (ZplankF1_t1 and ZplankR1_t1, see *Prosser, Martínez-Arce & Elías-Gutiérrez, 2013* for details). The reactions were cycled at 94 °C for 1 min, followed by five cycles of 94 °C for 40 s, 45 °C for 40 s and 72 °C for 1 min, followed by 35 cycles of 94 °C for 40 s, 51 °C for 40 s and 72 °C for 1 min, with a final extension of 72 °C for 5 min. PCR

products were visualized on a pre-cast 2% agarose gels (E-Gel© 96, Invitrogen, Waltham, MA, USA), and the most intense positive products were selected for sequencing.

## Sequencing and DNA barcode

Selected PCR products were sequenced using a modified (*Hajibabaei et al., 2005*) BigDye© Terminator v.3.1 Cycle Sequencing Kit (Applied Biosystem, Inc., Waltham, MA, USA), and then sequenced bidirectionally on an ABI 3730XL automated capillary sequencer using M13F and M13R sequence primers at the Biology Institute at the National Autonomous University of Mexico and at the Eurofins Genomics Louisville Laboratory. Sequences were edited using CodonCode© v 3.0.1 (CodonCode Corporation, Dedham, MA, USA) and uploaded to BOLD. In some cases, the original forward and reverse trace files uploaded to BOLD were checked again, consensus assembly was generated, and edited manually with Sequencher© 4.1.4. (Gene Codes Corporation, Ann Arbor, MI, USA), and then they were aligned using BioEdit© (*Hall, 1999*).

## Likelihood tree and distance analysis

COI sequences generated for *Holothuriophilus trapeziformis* in this study were compared with COI sequences from other pinnotherids collected in the Eastern Pacific coast of America, available in BOLD and/or GeneBank (Table S2). Sequence data, trace files, and primer details for all *H. trapeziformis* specimens and for the outgroup species are available under the dataset name "*Holothuriophilus trapeziformis* from Mexico" (DOI: dx.doi.org/10.5883/DS-PINMX1HT) in the Barcode of Life Data System (barcodinglife. org). Additionally, *H. trapeziformis* sequences were uploaded to GenBank (https://www. ncbi.nlm.nih.gov/genbank/). Accession numbers are noted in Table S2.

We calculated the best-fitting evolution model of nucleotide substitution for distance based on COI alignments for the Maximum Likelihood (ML) tree accordingly to the Akaike (AIC) and Bayesian (BIC) criterions (*Darriba et al., 2011*) using jModelTest© 2.1.10 (*Posada & Buckley, 2004*). The final tree was estimated with 1,000 bootstrap replicates in MEGA© 7.0 (*Tamura et al., 2013*). With the compress/expand feature of MEGA we prepared the final tree. Also, we estimated the interspecific COI genetic distances for the dataset using the Kimura-2 parameters distance method in MEGA. Values greater than 3% were considered the threshold for the delimitation of the species (*Hebert et al., 2003*).

Acronyms used in the text: BOLD, barcode of life database (boldsystems.org); BIN, barcode index number (*sensu Ratnasingham & Hebert, 2013*); BOLD-ID, Specimen ID in the Barcode of Life Data System; CNE-ICML-UNAM, National Collection of Echinoderms of the Institute of Marine Sciences and Limnology of the National Autonomous University of Mexico; DC-NHM, Division of Crustacea, Natural History Museum, Smithsonian Institution; SMF-ZMG, Senckenberg Museum für Naturkunde, Zoologisches Museum Göttingen University, Humboldt Universität, Berlin; UABC, Autonomous University of Baja California, Mexico; UMAR, Universidad del Mar campus Puerto Angel, Oaxaca, Mexico.

Collectors: AEV, Aidé Egremy Valdés; AGF, Andrea Glockner Fagetti; CCA, Carlos Cruz Antonio; AHM, Adanely Hernández Muñoz; FBV, Francisco Benítez Villalobos; FCC, Fernando Cortés Carrasco; HMC, Humberto Mesa Castillo; KFL, Karen Lizbeth Flores López; KMB, Karen Mesa Buendía; RGF, Rebeca Granja Fernández; VCH, Valeria Chavez García.

For this study, we obtained a field collecting permit for non-commercial scientific research purposes from Secretaría de Agricultura, Ganadería, Desarrollo Rural, Pesca y Alimentación (SAGARPA) and Comisión Nacional de Acuacultura y Pesca (CONAPESCA) (Collecting permit: PPF/DGOPA-301/17).

## RESULTS

We analyzed the morphology of 55 specimens (Table S1) from two coastal regions (northern and southern) in the Mexican Pacific (Fig. 1). Notable variations were determined after detailed morphological examination of the specimens. In particular, the morphology of the topotypes (northern region; Fig. 1A) shows variation in the general carapace shape outline and general appearance which looks stouter and eroded in contrast to that of the southern region.

Nevertheless, in all the specimens we observed diagnostic characters of the species as the subrectangular carapace shape, with its crested lateral margin, the compressed walking legs, and the gap between the fingers of the chelae. Also, the male first and second gonopods plus the DNA data, confirm that all material examined corresponds with *Holothuriophilus trapeziformis*.

**Systematics**
Infraorder Brachyura Latreille, 1802
Family Pinnotheridae De Haan, 1833
Genus *Holothuriophilus Nauck, 1880*

**Diagnosis** (modified from *Manning, 1993*). Carapace broader than longer, widest on mid anterior portion, transversely subcuadrangular, subrectangular, subovate or subtrapezoidal. Third maxilliped with ischium and merus fused, no suture line; exopod with one flagellar segment; palp with 3-segments; propodus conical, shorter than carpus, subspatuliform dactyl, articulated at basal of propodus, extending beyond end of propodus. Dactylus of walking legs short, similar and subequal. Abdomen of seven segments in both sexes.

### *Holothuriophilus trapeziformis Nauck, 1880*
(Figs. 2A–2G, 3A–3D, 4A–4K, 5A–5K, 6A–6D)

*Holothuriophilus trapeziformis Nauck, 1880*: 24, 66 [ovigerous female type]. —*De Man, 1887*: 721–722 [female (CW = 13.8 mm, CL = 10.5 mm)]. —*Manning, 1993*: 524–528, Fig. 3c [resurrected to *Holothuriophilus*]. —*Ng & Manning, 2003*: 903, 916–918, Figs. 7C–7F [designated female lectotype (CW = 7.7 mm, LC = 4.8 mm): SMF-ZMG

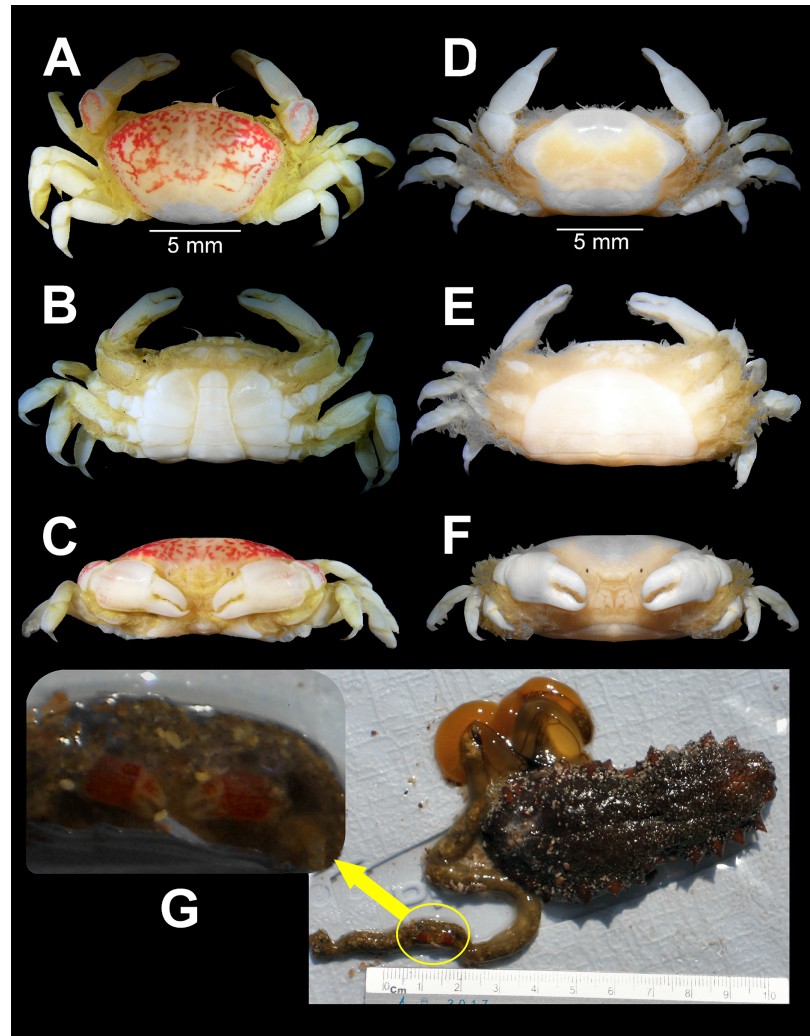

**Figure 2 *Holothuriophilus trapeziformis Nauck, 1880*.** (A–C) Male from Panteón Beach, Oaxaca, Mexico (UMAR-DECA-308): (A) dorsal view; (B) ventral view; (C) frontal view. (D–F) female from Agua Blanca Beach, Oaxaca, Mexico (UMAR-DECA-307): (D) dorsal view; (E) ventral view; (F) frontal view; (G) male inside the gut of *Holothuria* (*Halodeima*) *inornata*, from Pinitos Beach, Sinaloa, Mexico.

67/565a]. —*Ahyong & Ng, 2007*: 213–214, Fig. 20 [redescribed and refigured]. —*Campos, Peláez-Zárate & Solís-Marín, 2012*: 57–62, Figs. 1A, 1B, 2A–2D [female (CW = 9.1 mm, CL = 5.5 mm)].

*Pinnotheres trapeziformis*.—*Bürger, 1895*: 380–381, plate 9, Fig. 26, plate 10, Fig. 25 [female type (CW = 14 mm, CL = 10 mm), male (CW = 5 mm, CL = 8.5 mm)]. —*Adensamer, 1897*: 107. —*Schmitt, McCain & Davidson, 1973*: 5, 13, 89 [list].

*Pinnoteres trapeziformis Balss, 1957*: 1,417 [not 1956 *fide Schmitt, McCain & Davidson, 1973*].

*Pinnixa barnharti* (no *Rathbun, 1918*) *Caso, 1958*: 329; 1965: 254–26.

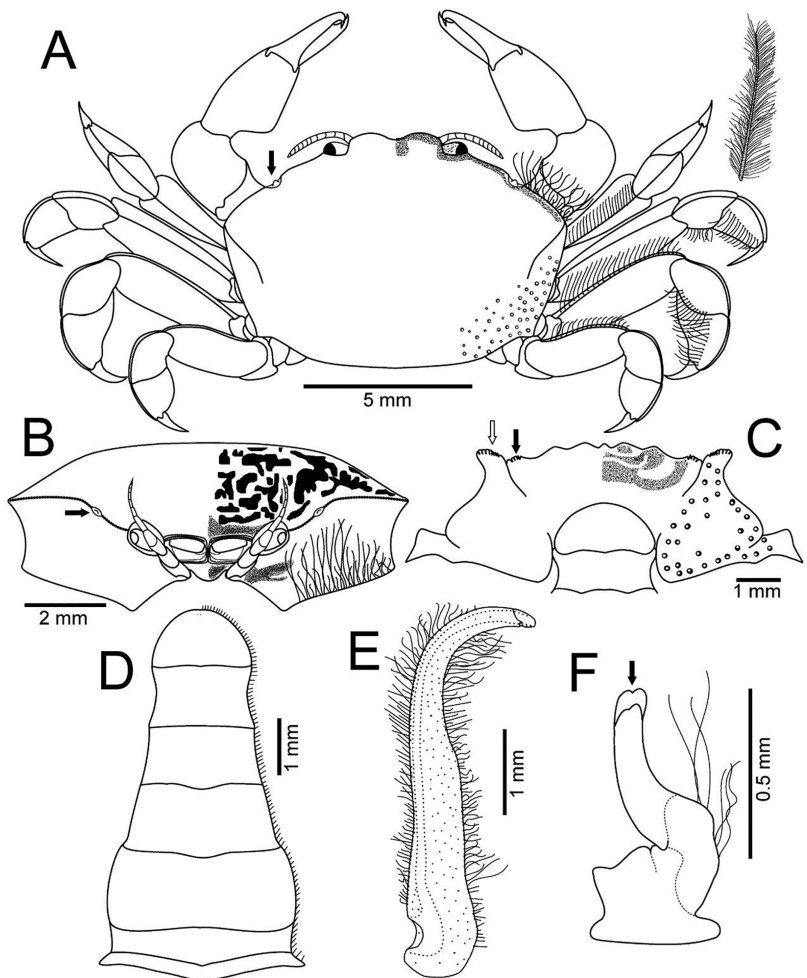

**Figure 3** *Holothuriophilus trapeziformis Nauck, 1880*. (A–D) Male from Panteón Beach, Oaxaca, Mexico (UMAR-DECA-308): (A) dorsal view (bold arrow indicates the internal blunt projection of the hepatic notch); (B) frontal view (bold arrow indicates the hepatic notch); (C) third-fourth sternal plate (bold arrow indicates the crenulated margin of the third plate, white arrow indicates the crenulated margin of the fourth plate); (D) abdomen; (E) abdominal view of the left first gonopod; (F) ventral view of the left second gonopod (bold arrow indicates the distal notch); (A) and (C) hollow circles indicating pits. Fine dots indicating pilosity. (A–D) Half of the illustration without ornamentation.

*Holothuriophilus* sp. *Campos, Díaz & Gamboa-Contreras, 1998*: 377, Fig. 1E.

**Material examined:** 56 specimens: 25 ovigerous females, 22 females, nine males (Table S1).

**General distribution:** Tropical Mexican Pacific.

**Previous records:** Mazatlan, Punta Tiburón (Sinaloa); Ixtapa (Guerrero).

**New records:** Pinitos Beach (Sinaloa); Nudista Beach, Zacatoso Beach, Caleta de Chón Beach (Guerrero); Agua Blanca Beach, Coral Beach, Camarón Beach, Panteón Beach, Estacahuite Beach, La Tijera Beach, San Agustín Beach, El Tejón Beach (Oaxaca).

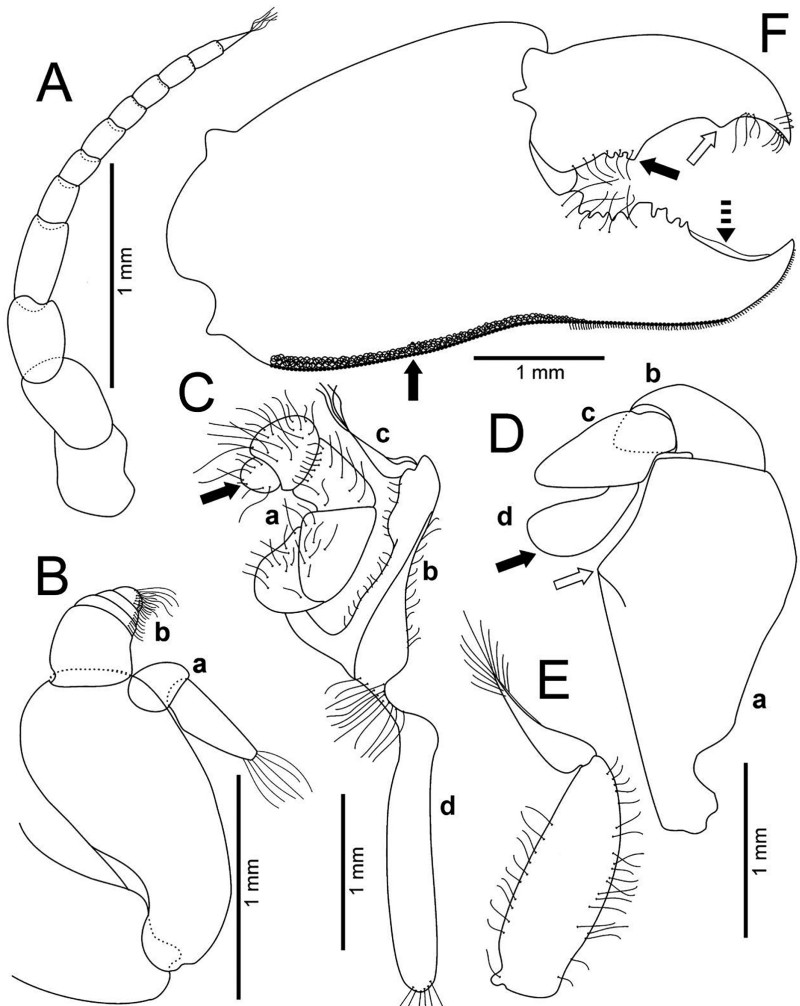

**Figure 4 *Holothuriophilus trapeziformis Nauck, 1880*.** (A–F) Male from Panteón Beach, Oaxaca, Mexico (UMAR-DECA-308): (A) antenna; (B) antennule: a, superior palp; b, inferior palp; (C) second maxilliped: a, endopod; b, exopod; c, exopod flagellum (bold arrow indicates the subrounded dactylus); (D) third maxilliped (setae not illustrated): a, ischiomerus (white arrow indicates the conspicuous projection); b, carpus; c, propodus; d, dactylus (bold arrow indicates the distal widened dactylus); (E) exopod of the third maxilliped; (F) chela (upper bold arrow indicating mid-posterior teeth and the lower one the granulated inferior margin; dashed bold arrow indicating the lamella; white arrow indicates the subdistal projection).

**BIN:** BOLD:ADE9974

**Carapace size (mm):** See Table S1.

**Diagnosis**. Carapace transversely subrectangular, suboval or subtrapezoidal. Crestated anterolateral margin, a hepatic notch with a blunt tooth inside. Inner surface of merus and carpus of chelipeds densely setose; ventral inner margin of the propodus with a row of short setae, without a gap when the cutting edge of propodus and dactylus meet; cutting edge of the dactylus with proximal denticles, a conspicuous medial tooth, and a distal convex or acute projection. Dorsal surface of the merus of WL1, WL 3 and WL 4 with setae, WL2 without seta. Abdomen with six somites plus free telson; on male,

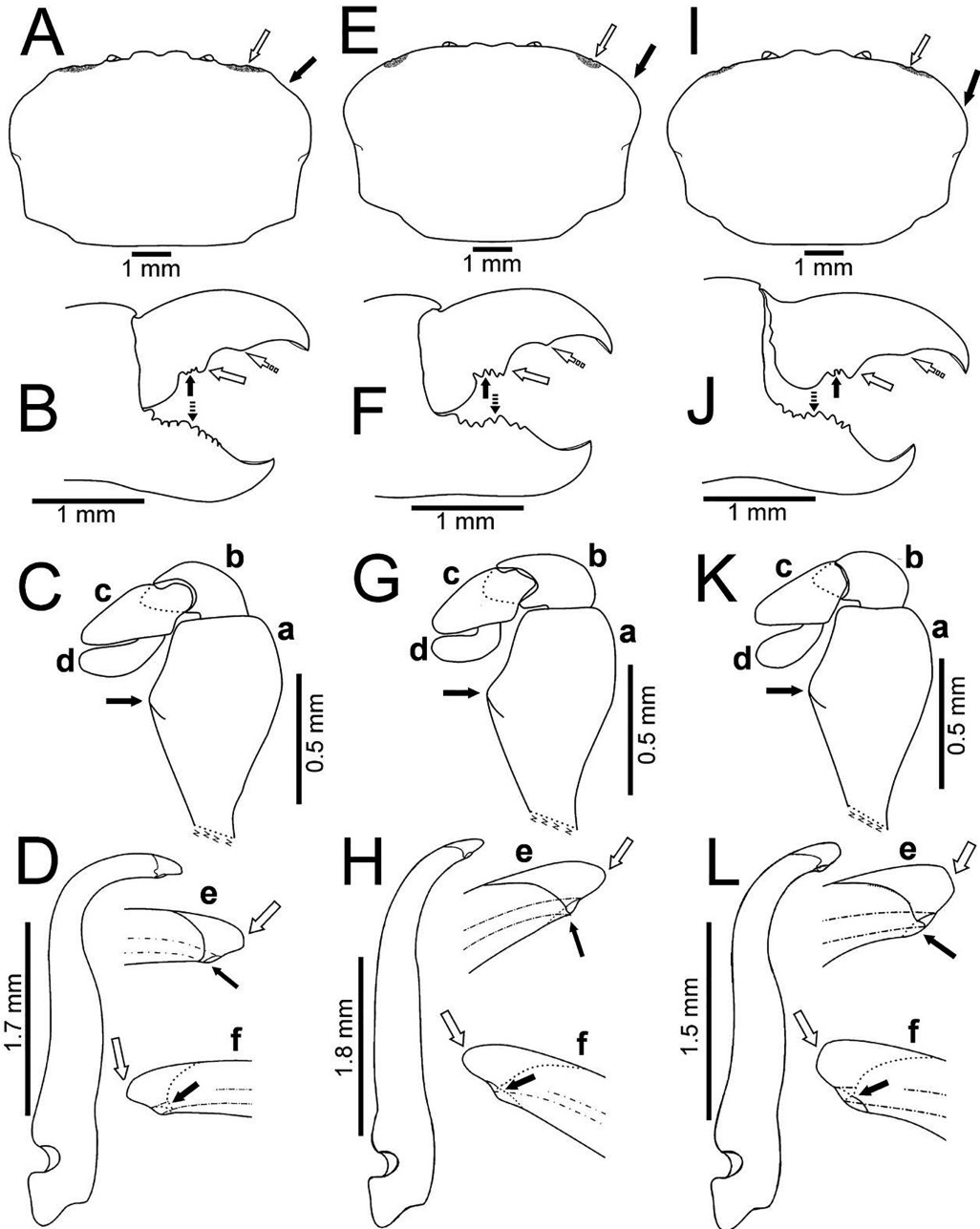

**Figure 5 Comparison between males of *Holothuriophilus trapeziformis Nauck, 1880* from the Pacific coast of Mexico.** (A–D) Sinaloa (DECA-1190; CW = 8 mm); (E–H), Guerrero (DECA-1148; CW = 8 mm); (I)–(L) Oaxaca (DECA-1270; CW = 8 mm). (A, E & I) Carapace outline (white arrow indicates the hepatic notch, bold arrow indicates the lateral lobes); (B, F and J) right chela, external view (bold arrow indicates the proximal teeth, white arrow indicates the medial tooth, dashed white arrow indicates the distal projection; dashed bold arrow indicates the medial tooth); (C, G & K) left Mxp3 endopod, external view (bold arrow indicates the ischiomerus projection); (D, H & L) first gonopod in ventral view; e, gonopod tip in ventral view; f, gonopod tip, dorsal view (e–f, white arrow indicates the truncated or acute distal margin and bold arrow indicates the ventral process).

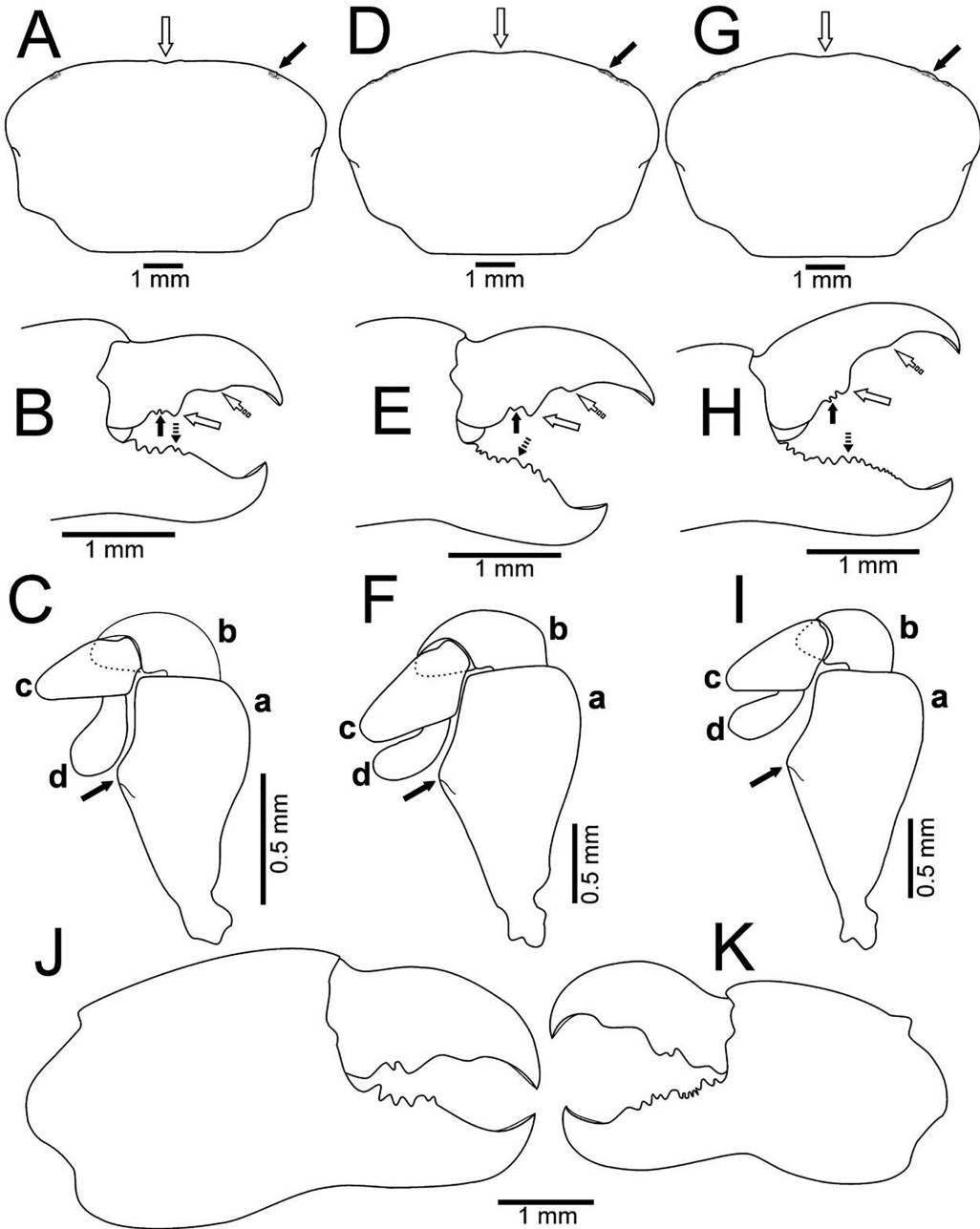

**Figure 6 Comparison between ovigerous females of *Holothuriophilus trapeziformis Nauck, 1880* from the Pacific coast of Mexico.** (A–C), Sinaloa (UMAR-DECA-1192; CW = 8 mm); (D–F) Guerrero (DECA-1149; CW = 8 mm); (G–I) Oaxaca (UMAR-DECA-1182; CW = 8 mm); (J & K) chelae, external view, Oaxaca (UMAR-DECA-1172; CW = 9 mm). (A, D & G) Carapace outline (white arrow indicates the frontal notch, bold arrow indicates the hepatic notch); (B, E & H) right chela, external view (bold arrow indicates the proximal teeth, white arrow indicates the medial tooth, dashed white arrow indicates the distal projection; dashed bold arrow indicates the medial tooth); (C, F & I) left Mxp3 endopod, external view (bold arrow indicates the ichiomerus projection).

margin of somite 4 to 6 concave, telson subrounded. First gonopod notably curved outward from its mid-distal portion.

**Description: Male (Figs. 2A–2C; UMAR-DECA-308; CW = 11 mm, CL = 7 mm):** Carapace, transversely subtrapezoidal, wider than long, CW/CL ratio ca. 1.6, mid-anterior portion widest; anterolateral margins slightly projected, crested, a hepatic notch with a blunt middle tooth (Figs. 3A, 3B; bold arrow); dorsal surface convex, smooth, undefined regions; mid-posterior and posterolateral surface with microscopic pits with variable pilosity and size (Figs. 3A); inferior lateral margin of the carapace with abundant plumose setae (Fig. 3A).

Front bilobed, scarcely visible in dorsal view, margin granulated, surface slightly pubescent (Fig. 3B).

Orbits small; eyes pigmented, same size as the orbits; ocular peduncle scarcely pubescent.

Antennules robust; peduncle with two segments, biflagellate, transversely folded inside fossae; upper flagellum with two articles, second article the longest and sharpened distally with apical setae (Fig. 4Ba); lower flagellum conic with four articles decreasing in size, the first three articles with a transverse line of simple setae, fourth article with two transverse lines of simple seta (Fig. 4Bb).

Antennae long, slender, with 12 articles, last article with short apical setae (Fig. 4A).

Pterygostomian region pubescence (Fig. 3B). Buccal frame trapezoidal, completely covered by Mxp3. Mxp2 endopod with five articles, with setae (Fig. 4Ca), dactylus subrounded and shorter than propodus (Fig. 4C); exopod with an article, wider distally, external surface with an elevated ridge (Fig. 4Cb), flagellum with long apical setae (Fig. 4Cc), epipodite long, distal margin rounded (Fig. 4Cd). Mxp3 ischiomerus fused without suture line, width/length ratio 0.7, mesial margin convex with setae, medial margin with a conspicuous projection (Fig. 4Da); carpus subconical, with short setae; propodus subconical (Fig. 4Dc); subspatuliform dactyl, widens distally (Fig. 4D), exceeds the length of the propodus, external surface with short plumose setae, external margin with long plumose setae; exopod with one article and a flagellum, external margin and external surface with short simple setae, slender flagellum, with plumose long setae (Fig. 4E).

Third sternal plate with anterior margin sinuous, anterolateral angles with crenulated margin (Fig. 3C), surface scarcely pilose (Fig. 3C); fourth plate slightly globose, surface with microscopic pits (Fig. 3C), anteroexternal angle curved outward, with margin crenulated (Fig. 3C).

Chelipeds subequal (Fig. 2A); external and lower surface of merus and lower margin of carpus with plumose setae (Figs. 2B); chelae wide with ventral margin microscopically granulated (Figs. 4F, 7C), dorsal margin slightly crested and bent inwards; fingers wider than longer, spooned with acute tip subdistally (Fig. 4F), without a gap where the cutting edge of propodus and dactyl meet (Fig. 7C). Movable finger shorter than fixed finger, crossed inward when closed. Movable fingers of both chelae with cutting edge sinuous, three medial teeth (Fig. 4F) and a subdistal convex projection (Fig. 4F). Fixed finger cutting

edge of both chelae with nine teeth and subdistal surface flatted (Fig. 4F), ventral margin with short setae.

Walking legs similar in both sides of the body, relative length WL3 > WL2 > WL1 > WL4, segments short, robust, compressed and dorsal margin crested, ventral surface with plumose setae; dorsal margin of merus of the WL1, WL3, WL4 with plumose setae, and WL2 naked; dactylus with acute tips, curved, and stout; length of the dactylus subequal to propodus of the WL1–WL3, and WL4 shorter than propodus (Fig. 3A).

Abdomen symmetrical, subtriangular, six free somites and telson margin with short setae, lateral margin from somites 4–6 slightly concave and narrowing, telson subrounded (Fig. 3D).

First gonopod slender, margins sinuous, mid-distal portion notably curved outwards, surface with abundant plumose setae (Fig. 3E). Second gonopod small, slightly bent inwards, tip pointing upwards, distal margins convex with shallow notch (Fig. 3F).

**Female (Figs. 2D–2F; UMAR-DECA-307; CW = 10.50, CL = 7):** Same as the male but with less abundant setae in the pterygostomian region and ventral margin of the chelae; dorsal surface of the merus of the WL1–4 and inner surface of the merus and carpus of the cheliped with long and abundant setae. Carapace slightly more convex. Abdomen subovate. See variation section for more details.

**Color in life:** Body beige or creamy white, dorsal surface of carapace and chelipeds carpus, and on the external surface of the chelae with red patches. In fixed and preserved specimens this pattern of color remains, or it could change from red to light or dark brown (Figs. 2A–2F).

**Habitat:** Marine. Associated with the sea cucumber *Holothuria* (*Halodeima*) *inornata*, living in its coelom and inside its intestine (Fig. 2G). This holothurian inhabits rocky-sand sea floor in shallow waters (0–18 m).

**Variation:** Morphological variation observed between the 8 mm individuals from the two regions (northern and southern) showed three general carapace shapes: subrectangular, suboval and subtrapezoidal.

The subrectangular carapace (Figs. 5A and 6A) was observed in 33% of the males and in 11% of females (three and five specimens, from the northern region). It has projected and straight margin frontal lobes, and straight anterolateral margin with a deep hepatic notch, eroded, and extended over the carapace (Fig. 5A) in males but in females it is less conspicuous (Fig. 6A). Males have truncated and scarcely projected anterolateral margins in which the anterior portion is concave (Fig. 5A) compared to straight in the females (Fig. 4A).

The subovate carapace (Figs. 5E, 5I, 6D, 6G) was observed in 56% of the males and in 85% of females (five and 40 specimens from the southern region). It has oblique and scarcely projected frontal lobes; anterolateral margin continues smoothly to the lateral margin forming a convex lobe (Figs. 5E, 5I), the hepatic notch in the males is deep, eroded

and extended (Figs. 5E, 5I), while in the females is shallow, slightly eroded, and less extended over the carapace (Figs. 6D, 6G).

The subtrapezoidal carapace (Figs. 2D, 7A) is present in 11% of the males and in 2% of females (one and two specimens from the southern region). It has scarcely projected frontal lobes, oblique and straight anterolateral margin ending lateral lobes.

Regardless of the carapace shape, the females have a more convex shell and margin of the frontal lobes and the eyes are not visible in dorsal view, and only a slight hepatic notch can be seen (Figs. 6A, 6D, 6G). This shape was more frequent in ovigerous females (10 specimens, 67%) than in non-ovigerous (five specimens, 33%).

Despite the variation in the carapace shape of both sexes, all the ratios (v.g. CW/CL ratio, length between the notch of the frontal lobes to the external orbital angle, length between the external orbital angle to the anterolateral angle, ishiomerus width/length ratio of the Mxp3) were constant.

Regardless of the shape of the carapace, the Mxp3 features show very subtle variations resulting from mounting and drawing. However, the observed variations are: The ischiomerus inner margin has a blunt or slightly acute projection (Figs. 5Ca, 5Ga, 5Ka, 6Ca, 6Fa, 6Ia). The dorsal margin of the carpus from convex (Figs. 5Kb, 6Cb, 6Fb, 6Ib) to straight (Figs. 5Cb, 5Gb). Distally expanded subspatulated dactylus (Figs. 5Kd, 6Cd) or narrower suboblong one (Figs. 5Cd, 5Gd, 6Fd, 6Id). However, in all cases there is a projected ridge on the internal surface of the carpus with setae, and the distal margin of dactylus slightly overreaches the propodus (Figs. 5Cd, 5Gd, 5Kd, 6Cd, 6Fd, 6Id).

The chelae fingers ornamentation is most variable and does not show a relationship with carapace shape or size. In both sexes the cutting edge of the movable finger has two or three proximal teeth (Figs. 5B, 5F, 5J, 6B, 6E, 6H). These teeth could be blunt or acute, but in the females, they always are acute. The medial tooth in males could be simple (Figs. 5B, 5J) or bicuspid (Fig. 5F), but in females it is always simple (Figs. 6B, 6E, 6H). The subdistal projection in both sexes can be blunt (Figs. 5B, 5J, 6B, 6H) or acute (Figs. 5F, 6E). The cutting edge of the fixed finger is more variable between sexes. In males it has six to nine proximal teeth (Figs. 5B, 5F, 5J), but in females it has 4 to 13 (Figs. 6B, 6E, 6H). In males it has one middle tooth which is always bicuspid (Figs. 5B, 5F, 5J) but in females could be bicuspid (Figs. 6B, 6E) or simple (Fig. 6H). Only one specimen (DECA-1172) had different sized chelae and a different teething pattern on the cutting edge of the fixed finger (Figs. 6J, 6K).

The first gonopod is related to the carapace shape. In caudal view it is similar between males of the northern region with a subrectangular carapace and that of the southern region of Oaxaca with suboavate and subtrapezoidal carapace. In these, the external and internal margins are sinuous (Figs. 5D, 5L), the curvature degree is approximately 90° (Fig. 5D) and 75° (Fig. 5L). The tip of the external margin is truncated (Figs. 5De–5Df, 5Le–5Lf), and the ventral margin of the tip has a blunt projection (Figs. 5De, 5Le).

Instead, other males from Guerrero, in the southern regions, with a suboavate carapace, have the external and internal margins of the first gonopod less sinuous and the curvature degree is approximately 65° (Fig. 5H). The tip of the external margin is convex (Figs. 5He–5Hf), and the ventral margin of the tip has an acute projection (Fig. 5He).

**Remarks:** The taxonomic history of *Holothuriophilus trapeziformis* was synthetized by *Campos, Peláez-Zárate & Solís-Marín (2012)*, and they highlighted the fact that the specimen identified by *Bürger (1895)* as a male based on the shape of the abdomen is actually a female. We observed the same situation in several young individuals. The presence of pleopods in all of the abdominal somites confirms that they are also females. In juvenile males the lateral margins of the abdomen are straight instead of concave and the first and second gonopods are present. This finding allows us to present the complete male morphology of *H. trapeziformis*.

All the biological material examined shows phenotypic variation, particularly between the individuals from the type locality in Mazatlan (northern region) with respect to those from of Guerrero and Oaxaca (southern region), but COI gene shows no differences.

With our detailed description of the male morphology, it is now possible to differentiate *Holothuriophilus trapeziformis* from *H. pacificus* with certainty by the carapace shape and CW/CL ratio. In *H. trapeziformis*, it can be subrectangular (Figs. 5A, 6A; CW/CL = ~1.6), suboval (Figs. 5E, 5I, 6D, 6G; CW/CL = ~1.6) or subtrapezoidal (Figs. 1A, 1D, 2A, 7A; CW/CL = ~1.6), while in *H. pacificus* it is subcuadrangular (Fig. 7E; CW/CL = ~1.2).

Nevertheless, the mxp3 does not present differences between both species, except for the flagellum of the exopod that according to the illustrations is robust (Figs. 7B, 7J, 7K, 8A) in *H. trapeziformis*, and slender in *H. pacificus* (Figs. 7F, 8D).

The abdomen of *Holothuriophilus trapeziformis* is subtriangular in males, with lateral margins narrowing from the fourth to the sixth somite. The third somite has convex lateral margins. The sixth somite has concave lateral margins, and the telson is subrounded and wider than long (Fig. 8B). In *H. pacificus*, it is triangular, the lateral margins are almost straight. The third and sixth somite lateral margins are concave, and the telson is subtriangular and more extended than wide (Fig. 8E).

In the case of adult ovigerous and non-ovigerous females of *Holothuriophilus trapeziformis*, the abdomen is suboval and wider than long. The first somite has convex lateral margins. The second somite has sinuous distal margins. The third somite has oblique and downward lateral margins. The sixth somite has oblique and outward lateral margins, and the telson has a length to width ratio ca. 0.2 (Fig. 7D). In contrast, in *H. pacificus* it is suboval and longer than wide. The first somite has concave lateral margins. The second somite has almost straight distal margins. The third somite has oblique and upwards lateral margins. The sixth somite has convex lateral margins, and the telson has a length to width ratio ca. 0.3 (Fig. 7H).

The first gonopod of *Holothuriophilus trapeziformis* has a sinuous lateral margin with a larger distal portion curved outwards, with abundant setae (Fig. 8C). In *H. pacificus* it is straight, with the distal portion slightly curved outwards, and less setae (Fig. 8F).

**Distribution and ecological comments:** With this study we increased the previous known distribution range that went from Punta Tiburón (Sinaloa state) to Playa Tejón (Oaxaca state). We found crabs in the coelom cavity and near the cloaca of the host, as mentioned by *Manning (1993)*, *Campos, Peláez-Zárate & Solís-Marín (2012)*, and

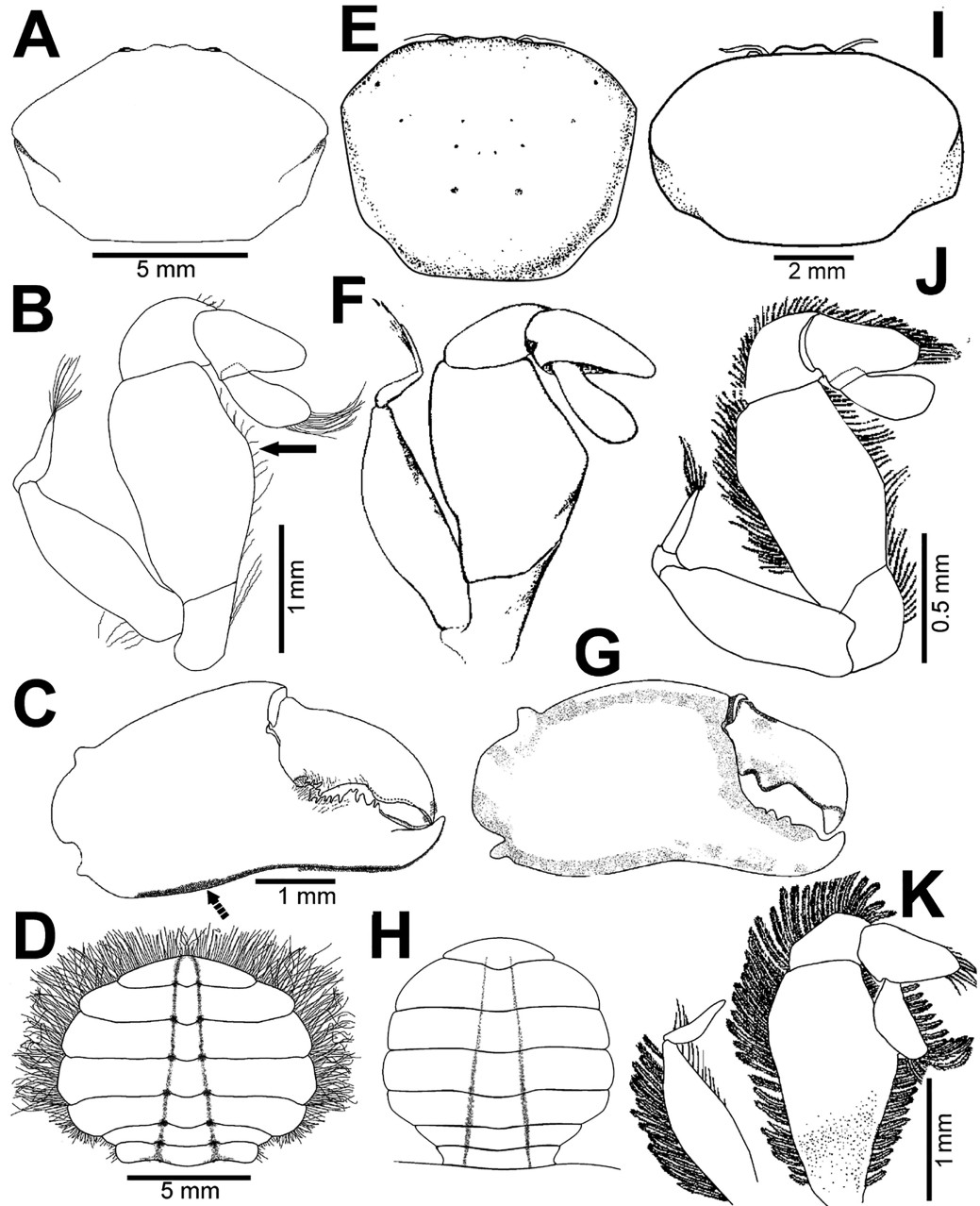

**Figure 7 Comparison between females: *Holothuriophilus trapeziformis *Nauck, 1880** and *H. pacificus* (*Poeppig, 1836*). (A–D) *H. trapeziformis* from Camarón Beach, Oaxaca, Mexico (UMAR-DECA-1163): (A) carapace; (B) third maxilliped (bold arrow indicates the ischiomerus projection); (C) chela (dashed bold arrow indicates the inferior granulated margin); (D) ovigerous abdomen; (E–H) *H. pacificus* from San Vicente, Chile (drawing after *Garth, 1957*: figs. 10E, F, G, H as *Pinnaxodes silvestrii*): (E) carapace; (F) third maxilliped; (G) chela; (H) abdomen; (I and J) lectotype of *H. trapeziformis* from Mazatlan, Mexico (drawing after *Ahyong & Ng, 2007*: Figs. 20A, C) (I) dorsal view of carapace; (J) third maxilliped; (K) *H. trapeziformis* from Guerrero, third maxilliped of the adult female of (drawing after *Campos, Peláez-Zárate & Solís-Marín, 2012*: Figs. 2B, C). Scale of E = x3.5, F = x18.6, G = x4.6, H = x2.9 (*fide Garth, 1957*).

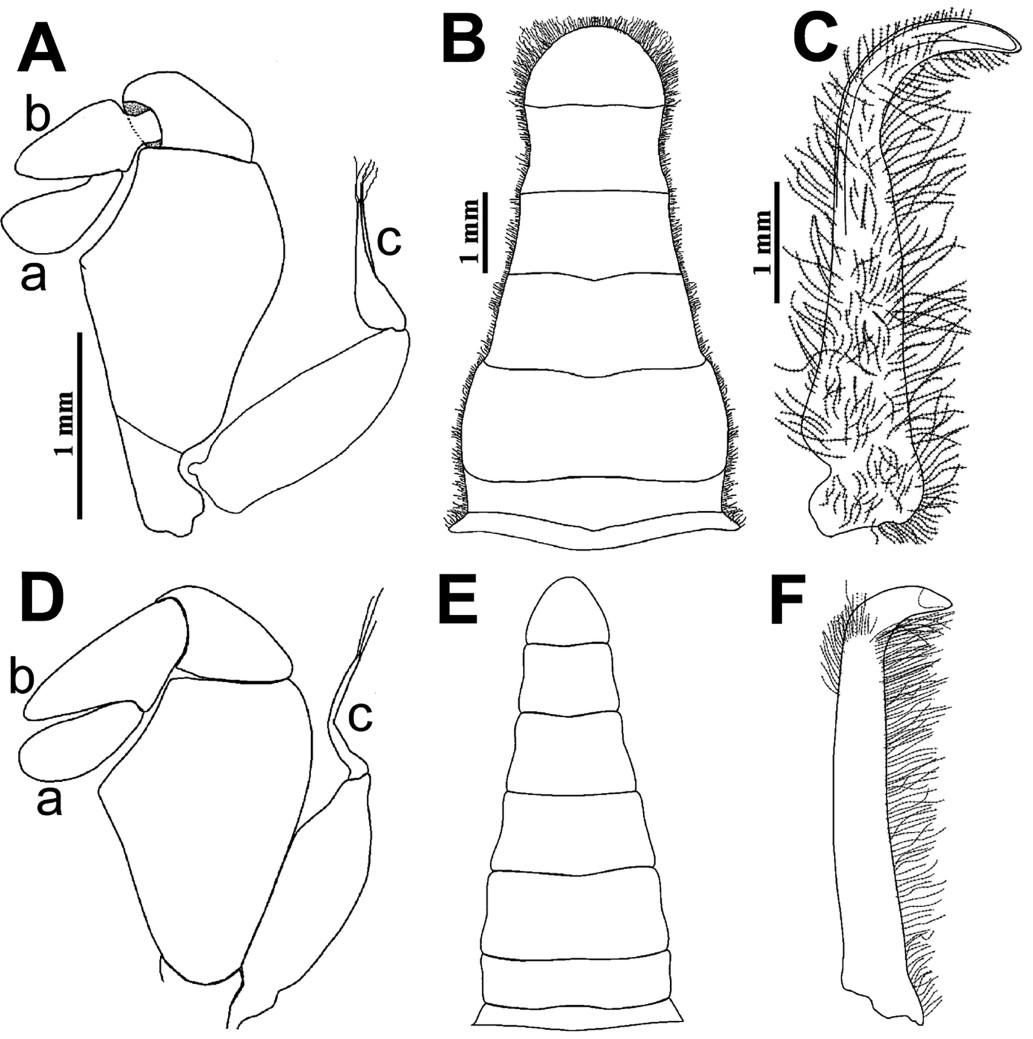

**Figure 8 Comparison between males: *Holothuriophilus trapeziformis Nauck, 1880* and *H. pacificus* (*Poeppig, 1836*).** (A–C) From Panteón Beach, Oaxaca, Mexico; (C) ventral view of the first gonopod, Mexico (UMAR-DECA-308). (D–F) From Talcahuano, Chile (drawing after *Garth, 1957*: Figs. 11A, B, C). (A & D) Third maxilliped; a, dactylus; b, propodus; c, exopod flagellum. (B & F) Abdomen. (C & F) First gonopod. Scale of D = x21, E = x6, F = x25 (*fide Garth, 1957*).

*Honey-Escandón & Solís-Marín (2018).* For the first time, we found the crab within the intestine (Fig. 1G).

*Holothuria* (*Halodeima*) *inornata* is distributed throughout the Tropical Eastern Pacific from the Gulf of California, Mexico to Ecuador, and in the temperate island Lobos de Afuera, Peru (*Prieto-Rios et al., 2014*; *Honey-Escandón & Solís-Marín, 2018*). It also represents an important fishery resource throughout its distribution range (*Santos-Beltrán & Salazar-Silva, 2011*). There are no records for *Holothuriophilus trapeziformis* outside the Pacific coast of Mexico.

**DNA Barcodes**

From the 56 crabs examined (Table S1), 51 were processed. The number of base pairs was between 549 bp and 648 bp for 37 specimens with a sole Barcode Index Number (BIN; *Ratnasingham & Hebert, 2013*) in the BOLD database: ADE9974. Of those, 35 produced high quality barcodes. The 14 crabs that could not be amplified correspond to old museum material and recent collections that were not fixed according to the *Elías-Gutiérrez et al. (2018)* protocol. A BLAST query in GenBank confirmed our sequences to belong to a brachyuran lineage.

**Maximum likelihood tree and genetic distance analysis**

The best nucleotide substitution model according to the AIC and BIC criterion was General Time Reversible under a gamma distribution (GTR+G) model (*Nei & Kumar, 2000*). The Maximum-Likelihood (ML) distance method under the selected model delimited the 37 sequences of *Holothuriophilus trapeziformis* from the dataset (DS-PINMX1HT) in a single cluster. The cluster of *H. trapeziformis* is well separated from *H. pacificus* as shown in Fig. 9, with a 12% to 14% divergence among all specimens. *Holothuriophilus* is also close to the *Calyptraeotheres* clade, but far from other species (Fig. 9) with an interspecific divergence ranging from 12% to 19%. The intraspecific divergences in *H. trapeziformis* ranged from 0 to 2.2%. This result is congruent with the BOLD distance summary analyses, which shows an average distance of 0.73% and a maximum of 2.27% for sequences with more than 500 bp.

## DISCUSSION

A crucial problem for traditional taxonomy based solely on morphology, is the variability of the phenotype of decapods. In pinnotherid taxonomy, a crucial goal is to provide a complete description of the species with detailed illustrations of common and unusual structures for comparative purposes (*Derby & Antema, 1980*; *Ahyong, Komai & Watanabe, 2012*; *Salgado-Barragán, 2015*). In that regard, characters previously not described like the antenna, the antennule, the Mxp2, and the second male gonopod show no differences between all the material examined, despite the variations noted above. These variations are greater when comparing individuals from the northern region of Mexico (which include the topotype in Mazatlan, Sinaloa state) to those from the southern region (Guerrero and Oaxaca states). However, COI data analysis confirms all the material examined corresponds to the same species based on the thresholds to delimitate species proposed by *Hebert et al. (2003)* and *Lefébure et al. (2006)*.

Phenotype variation is the result of a plastic response to different environmental pressures (*Hurtado, Mateos & Santamaria, 2010*; *Rossi & Mantelatto, 2013*) or due to recent or historical processes that limit the flow of genes because of environmental barriers (*Wares, Gaines & Cunningham, 2001*; *Avise, 2009*). Despite the fact that these processes are well documented, in the case of brachyuran crabs, there is evidence showing that it does not occur in grapsids (*Cassone & Boulding, 2006*), ocypodids (*Laurenzano, Mantelatto & Schubart, 2013*), pinnotherids (*Ocampo et al., 2013*), sesarmids (*Zhou et al., 2015*), and varunids (*Zhang et al., 2017*). However, for pinnotherids, the several

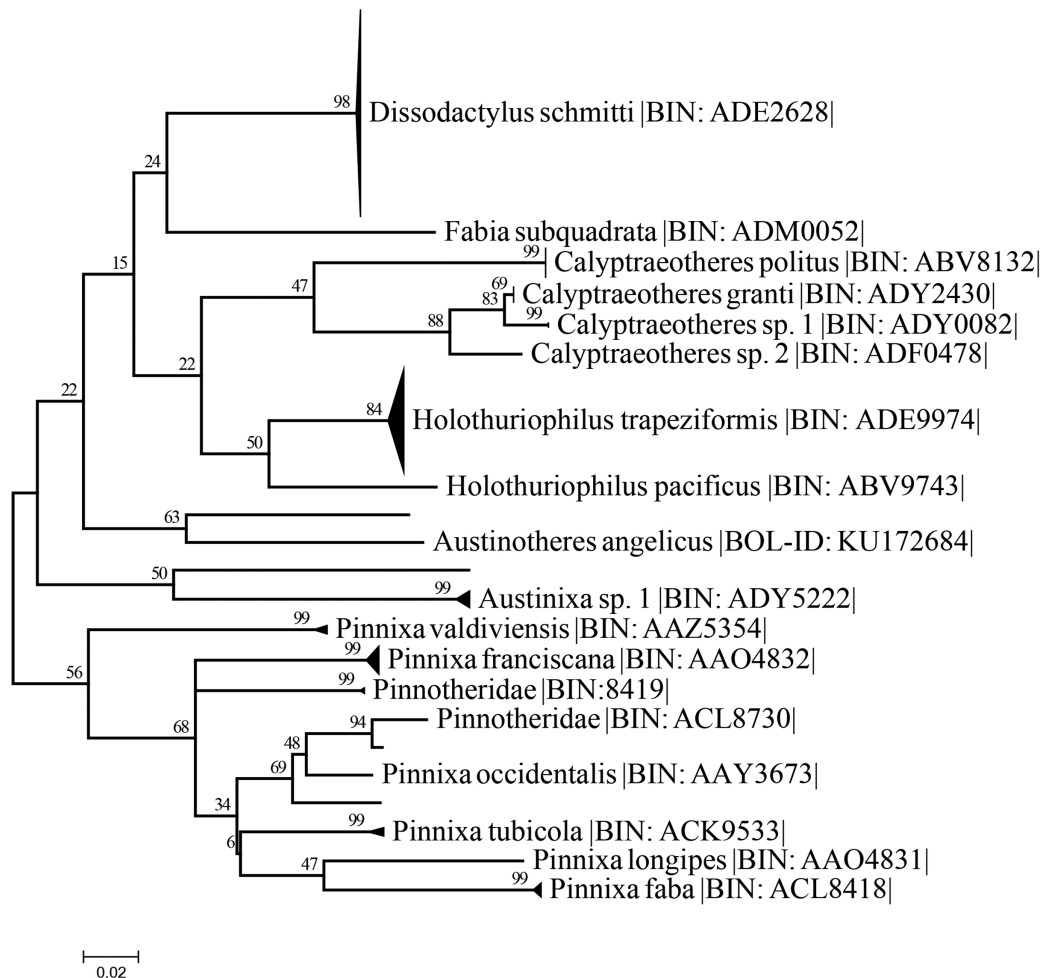

**Figure 9 Condensed unrooted Maximum likelihood tree based on mitochondrial cytochrome c oxidase (COI) with the General Time Reversible with gamma distribution (GTR+G) model.** Data: BOLD process ID, species name, associated BIN. Branch values represent bootstrap probabilities (1,000 permutations).

long-lasting growth phases require a specific host or a variety of hosts to complete them (*Bousquette, 1980*; *Hamel, Ng & Mercier, 1999*; *Ocampo et al., 2011*) and represent another drawback. Nevertheless, it allows them to maintain connectivity between populations throughout their geographical distribution range (*Haines, Edmunds & Pewsey, 1994*; *Hamel, Ng & Mercier, 1999*; *Ocampo et al., 2012*, *2013*; *Guilherme, Brustolin & de Bueno, 2015*; *Becker & Türkay, 2017*).

In the case of *Holothuriophilus trapeziformis*, it is considered a specific endobiontic parasite of its host (*Nauck, 1880*; *Campos, Peláez-Zárate & Solís-Marín, 2012*), resulting in possibly limited connectivity through larval dispersal. In addition to the above, the particular oceanographic conditions known along the Pacific coast of Mexico and the distribution of the host (*Hurtado et al., 2007*; *Paz-García et al., 2012*; *Prieto-Rios et al., 2014*; *Gómez-Valdivia, Parés-Sierra & Flores-Morales, 2015*; *Honey-Escandón &*

*Solís-Marín, 2018*), could explain the morphological differences observed between the northern specimens compared to those of the south.

Currently, with the complete description of the male using the new characters described here, we can conclude that *Holothuriophilus trapeziformis* is different from *H. pacificus*.

Regarding the DNA barcoding approach for the COI gene, in a difficult group to work with, we were successful thanks to the injection of ethanol inside the body of the crabs through the joints of the exoskeleton, and the use of semi-degenerate zooplankton primers (*Prosser, Martínez-Arce & Elías-Gutiérrez, 2013*) instead of Folmer's (*Mantelatto et al., 2016*). We obtained the amplification of 72% of the specimens and 69% sequencing success.

The resulting maximum likelihood tree allowed us to confirm *Holothuriphilus trapeziformis* as a separate species, indicating a divergence from 12% to 14% against the closest taxa, *H. pacificus*. Also, our tree agrees with *Palacios-Theil, Cuesta & Felder (2016)* regarding the association of the genus *Holothuriophilus* and *Calyptraeotheres*.

The taxonomic status of *Holothuriophilus trapeziformis* is now complete, based on the morphology of both sexes, their distribution, specificity to a single host, and the DNA barcodes. We believe that *Holothuriophilus trapeziformis* with its host reflects the restricted habitat in which it lives and possibly the local environmental barriers in the Pacific coast of the American continent.

## ACKNOWLEDGEMENTS

We are grateful to the Chetumal Node of the Mexican Barcode of Life (MEXBOLD) for their support for the genetic analysis, in particular to Alma Estrella García-Morales who assisted with the DNA process of the biological samples. J. Rolando Bastida-Zavala gave us access to the collection material of Laboratorio de Sistemática de Invertebrados Marinos (LABSIM) from Universidad del Mar. Fernando Álvarez-Noguera and José Luis Villalobos-Hiriart provided access to the collection material of the Colección Nacional de Crustáceos (CNCR) del Instituto de Biología de la Universidad Nacional Autónoma de México. Virgilio António Pérez and staff from Buceo Huatulco supported our field work. Miriam Steinitz-Kannan from Northern Kentucky University kindly assisted us with the English edition of this manuscript.

### Funding

Fernando Cortés-Carrasco received a fellowship from the National Council of Science and Technology (CONACYT). The funders had no role in study design, data collection and analysis, decision to publish, or preparation of the manuscript.

### Grant Disclosures

The following grant information was disclosed by the authors:
National Council of Science and Technology (CONACYT).

## Competing Interests

The authors declare that they have no competing interests.

## Author Contributions

- Fernando Cortés-Carrasco conceived and designed the experiments, performed the experiments, analyzed the data, prepared figures and/or tables, authored or reviewed drafts of the paper, and approved the final draft.
- Manuel Elías-Gutiérrez conceived and designed the experiments, performed the experiments, analyzed the data, authored or reviewed drafts of the paper, and approved the final draft.
- María del Socorro García-Madrigal analyzed the data, prepared figures and/or tables, authored or reviewed drafts of the paper, and approved the final draft.

## Field Study Permissions

The following information was supplied relating to field study approvals (*i.e.*, approving body and any reference numbers):

Secretaría de Agricultura, Ganadería, Desarrollo Rural, Pesca y Alimentación (SAGARPA) and Comisión Nacional de Acuacultura y Pesca (CONAPESCA), permit PPF/DGOPA-301/17.

## DNA Deposition

The following information was supplied regarding the deposition of DNA sequences:

The DNA sequences are available at BOLD: PINMX1HT (DOI 10.5883/DS-PINMX1HT). The sequences are also available at GenBank: MW544291–MW544443.

## Supplemental Information

Supplemental information for this article can be found online at http://dx.doi.org/10.7717/peerj.12774#supplemental-information.

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
