# Peer review of "Holothuriophilus trapeziformis Nauck, 1880 (Decapoda: Pinnotheridae) from the Pacific coast of Mexico: taxonomic revision based on integrative taxonomy"

_PeerJ, doi:10.7717/peerj.12774_

## Round 0.1 · original submission · Major Revisions

I have now heard back from two reviewers, both of whom have suggested numerous edits to your work. In particular, reviewer #2 has extensive comments on the structure and the discussion of your paper, which you should strongly consider in any revision.

Reviewer 1 ·

Basic reporting

-Cortés-Carrasco et al. studied and re-described Holothuriophilus trapeziformis from Mexico's Pacific coast, especially for the male. The morphology characteristics that the authors provided in this manuscript are essential and well described for future research of H. trapeziformis. I assume we need more studies to confirm the relationship between these species and their host (sea cucumbers). However, the population genetics part of this study is fragile, and it will not add much. The phylogenetic tree and the morphological characters are much better.

-Editing of the English language is needed for this manuscript.

-There is no map of the sampling locations.

Experimental design

-The genetic part needs a lot of rearrangements and corrections. The authors studied the population genetic structure based on one unspecific marker, a low number of samples, and populations in a small area. Additionally, it is hard to followup some parts, such as showing four populations in Table 1, but it shows only three populations in Figure 9 (which one is correct). Also, it is not clear that they studied the genetics of sea cucumbers or no. Please delete the population genetic section from this study.

-The authors used a specific primer for the freshwater microcrustaceans (ZplankF1_t1 and ZplankR1_t1)

Validity of the findings

-The morphology characteristics that the authors provided in this manuscript are essential and well described for future research of H. trapeziformis.

-Please add the accession numbers of the sequences provided by GeneBank.

Additional comments

Some minor corrections (not all):
-The manuscript needs English language editing. Some phrases, articles, and punctuations are not needed, and some sentences are unclear or hard to follow, for instance:
-Line 37: please remove the phrase "which is".
-Line 39: please remove the article "the" from "the females".
-Line 41-43: the sentence "Our goal ……." is hard to follow.
-Line 45-48: the sentence "We also….." is unclear and hard to follow. Also, remove "," after compared in line 45.
-Please reconstruct/rearrange the abstract and remove the unnecessary information. In the results section in Line 49, the authors started with the introduction and methods again in this part. Also, the authors' goal in Line 41 has the same meaning in Line 44; please describe the methods used to study morphology and DNA.
-Line 41 and Line 59-60: please rephrase "…..for the first time."
-Line 66-67: The number of known species is 16, not 13. Ng & Manning (2003) mentioned (P:901, last paragraph) that 16 species are known to be parasites of sea cucumbers; 13 of them occur in the Indo-West Pacific.
-Line 70: Add a reference to support this information.
-Line 87-88: "based on a single morphological character" could you please let the readers know what that single morphological character is?
-Line 94: Nauck (1880) made a great effort to record the information at that time. Therefore, please delete this sentence or rewrite it "did not designate a holotype…..".
-Line 193-194: It is hard to followup! Are the host sea cucumbers sequenced? If so, how many specimens?
-Line 204: Zooplankton primers (ZplankF1_t1 and ZplankR1_t1) are specific primers for freshwater microcrustaceans. Are these primers used for Holothuriophilus trapeziformis and sea cucumbers? I assume there many specific primer sets for marine microcrustaceans.
Line 222: Are COI sequences from other pinnotherids collected in America's Eastern Pacific coast were collected in this study? If no please add the reference.
-Line 227: please add the accession numbers of uploaded data on GeneBank.

·

Basic reporting

Although this is a straightforward paper redescribing a single brachyuran species, Holothuriophilus trapeziformis Nauck, 1880, the manuscript is lengthy and difficult to trace what is the point of discussions.

Experimental design

The aims of the present study are not clearly stated, but if a part of them are redescription of the species and finding diagostic chracters from congeners, authors should be able to have enough dataset but not well presented.

Validity of the findings

Authors have good amount of data with good photographs and drawing.

Additional comments

Since authors have good amount of data and specimens, and it is important to redescribes the species in detailed, I encourage authors to revise the manuscript substantially and re-submit the ms to more specialized journal.

---

## Round 0.2 · Major Revisions

I have heard back from the same two reviewers as the last round of review. Both find the work greatly improved, but there are still some areas to address and room for improvement. Please do not forget to look at the attached PDF for comments as well.

Reviewer 1 ·

Basic reporting

Minor English editing such as:
-Line 117-122 long sentence, and it will lead to hard to follow.

-Adding comma in some sentences, for instance, before "and the male" line 87.

-Could you please check this reference (I think it is not cited in the text)? "Lefébure T, Douady CJ, Gouy M, Gibert J. 2006. Relationship between morphological and molecular divergence within Crustacea: Proposal of a molecular threshold to help species delimitation. Molecular Phylogenetics and Evolution 40(2006):435–447"

Experimental design

N/A

Validity of the findings

N/A

·

Basic reporting

The current version of the manuscript is well modified from the previous version, but there is much room to improve it. See comments in the attached ms.

Experimental design

Although this is a simple re-description paper, targets and aims of comparisons are unclear. Are there any null-hypotheses that particular groupings (e.g. locality, carapace shape) may represent different species? Since points in questions are not well raised, it is very difficult to follow the flow.

Validity of the findings

Further justification may be needed, for example genetic comparison between "groups" (e.g. locality, carapace type).

Additional comments

See attached file for further comments.

---

## Round 0.3 · Minor Revisions

While the former reviewer declined to review this round, from looking over your new version and responses I can see you have worked hard on this revision, and I consider it acceptable from a scientific point of view. However, the English here and there remains to be edited to some degree, preferably by a taxonomist with familiarity with crustaceans, and not a for-hire company, which often make taxonomic papers worse rather than better.

---

## Round 0.4 · Minor Revisions

Thank you for your revision. Unfortunately, the English is not yet acceptable, and there are a lot of areas that need editing. You can either ask a colleague familiar with crustacean taxonomy and fluent in English, or ask PeerJ or another professional editing service. I am sorry, but I cannot accept the paper until the English is up to international standard.

---

## Round 0.5 · accepted · Accept

Thank you for all of your hard work; I am now happy to move your paper intro production, and look forward to seeing the published version in the near future.